



# Quantifying location error to define uncertainty in volcanic mass flow hazard simulations

Stuart R. Mead[1], Jonathan Procter[1] and Gabor Kereszturi[1]

Volcanic Risk Solutions, Institute of Agriculture and Environment, Massey University, Palmerston North, New Zealand.

*Correspondence to:* Stuart R. Mead (S.Mead@massey.ac.nz)

**Abstract.** The use of mass flow simulations in volcanic hazard zonation and mapping is often limited by model complexity (i.e. uncertainty in correct values of model parameters), a lack of model uncertainty quantification, and limited approaches to incorporate this uncertainty into hazard maps. When quantified, mass flow simulation errors are typically evaluated on a pixel-pair basis, using the difference between simulated and observed ('actual') map-cell values to evaluate the performance of a model. However, these comparisons conflate location and quantification errors, neglecting possible spatial autocorrelation of evaluated errors. As a result, model performance assessments typically yield moderate accuracy values. In this paper, similarly moderate accuracy values were found in a performance assessment of three depth-averaged numerical models using the 2012 debris avalanche from the Upper Te Maari crater, Tongariro Volcano as a benchmark. To provide a fairer assessment of performance and evaluate spatial covariance of errors, we use a fuzzy set approach to indicate the proximity of similarly valued map cells. This 'fuzzification' of simulated results yields improvements in targeted performance metrics relative to a length scale parameter, at the expense of decreases in opposing metrics (e.g. less false negatives results in more false positives) and a reduction in resolution. The use of this approach to generate hazard zones incorporating the identified uncertainty and associated trade-offs is demonstrated, and indicates a potential use for informed stakeholders by reducing the complexity of uncertainty estimation and supporting decision making from simulated data.

## 1 Introduction

Mass flow numerical models are frequently used to predict the hazard from future events (e.g. Procter et al., 2010a;Scott et al., 2005;Scott et al., 1997;Aguilera et al., 2004;Pistolesi et al., 2014;Darnell et al., 2013;Thouret et al., 2013), understand fundamental processes within mass flows, investigate previous events (e.g. Iverson and George, 2016), and determine impacts to elements exposed to the flow (e.g. Zeng et al., 2015;Mead et al., 2017). Their utility and advancing computational power have positioned numerical models, of various scales and complexity, as a critical risk management and decision making tool (Bennett et al., 2013). An important element of risk management and decision making in a natural hazard context is the quantification and communication of uncertainty (Thompson et al., 2015;Doyle et al., 2014). In numerical modelling, much uncertainty is associated with a models predictive accuracy, where inaccuracies can stem from input and boundary conditions, model assumptions and numerical limitations. However, testing model accuracy, eliciting the effect of inputs, assumptions and limitations on accuracy and communicating these effects is a non-trivial task (Bennett et al., 2013;Jakeman et al., 2006;Wealands et al., 2005;McDougall, 2016).



The fundamental requirement for assessing model accuracy is the establishment of a baseline, 'true', dataset for comparison between model and reality. Experimental facilities and studies (e.g. Iverson et al., 2010;Lube et al., 2015) can provide detailed, scaled down observations of mass flow processes to validate, improve and develop numerical models. Some known, important processes (e.g. initial conditions, unconfined flow, interaction with

topography) are simplified in these facilities, and a complete assessment of model accuracy requires the application to real scale, 'real world' mass flows. However, such measurements of mass flow characteristics are mostly limited to post-hoc analyses of events where data are derived from static (i.e. single points of data not varying in time) observations such as deposit depth, flow outlines and flow height markers (Charbonnier et al., 2017;Charbonnier et al., 2013;Procter et al., 2014). Temporally-varying measurements of mass flows do exist, for

example when the occurrence is known *a priori* (e.g. in Procter et al., 2010b) or when permanent sensors (e.g. seismometers) capture some aspect of the flow (Velio et al., 2018;Walsh et al., 2016). While these direct observations can provide benchmarking opportunities for mass flow models, they are rare, and currently not comprehensive enough to quantify the entire range of mass flow sizes and behaviours. Therefore, performance measurement will predominantly need to utilise static, post-hoc observations.

In general, quantifications of model performance can be split into global or local comparisons (Wealands et al., 2005). Global comparisons characterise a mass flow into single, easy to interpret metrics (e.g. length of flow, area inundated; Charbonnier et al., 2017;Mergili et al., 2017) but can disguise both spatial and temporal divergent behaviour (Bennett et al., 2013). Local comparisons of model performance typically utilise a confusion matrix (Bennett et al., 2013;Charbonnier et al., 2017;Mergili et al., 2017) to classify proportions of correctly or

incorrectly simulated data points, where spatial accuracy of simulators can be evaluated by comparing pixel-pairs on a map (Wealands et al., 2005). However, there is no universal metric for quality (Bennett et al., 2013), and various measures can be used depending on objectives or potential uses of the simulator (Bennett et al., 2013;Jakeman et al., 2006). Pixel-pair comparisons are also prone to registration issues (e.g. where systematic errors in base data or observations shift results, Wealands et al., 2005;Koch et al., 2015;Foody, 2002) that, even

if small, can decrease overall accuracy metrics (see e.g. Charbonnier et al., 2017) due to the lack of tolerance for spatially or quantitatively minor errors. The conflation of quantity and spatial errors is particularly relevant for mass flow models, where the likelihood of errors generally decreases as flow depths increase.

The conflation of these scale and location errors and reliance on precise co-location in comparisons contrasts with human (i.e., qualitative) comparisons that provide for some error tolerance, and, through focusing on basic spatial

structure, logical coherence and importance-weighting of similarities (Hagen, 2003;Koch et al., 2015;Wealands et al., 2005). Human visual comparison is a powerful method for comparing and evaluating spatial field results (Wealands et al., 2005) and many comparison approaches attempt to emulate the human ability to distinguish between residual or random errors and errors in registration (i.e. account for co-location errors) and resolution (i.e. account for errors that are only significant at certain scales, Costanza, 1989). These approaches, identified

and reviewed in Wealands et al. (2005), include multi-resolution comparisons (Costanza, 1989) to identify similarity of measurements with scale; region clustering, segmentation and homogenisation to identify and compare patterns in the spatial field; importance weighting to focus performance evaluation on the most (hydrologically) important regions; and fuzzy comparisons to represent relative membership ('fuzziness') of each map cell to a certain category (e.g. inundated/not inundated). Despite this range of potential performance

evaluation methods, there is no universal criterion or method to evaluate model quality (Bennett et al., 2013) and



there are few examples of mass flow model performance evaluations that identify the level of both location and quantification error.

Robust, objective and complete evaluations of model performance that quantify the uncertainty of predictions and effect of model and input parameters are essential for the development and use of mass flow models as a reliable hazard forecasting tool (McDougall, 2016;Calder et al., 2015). The selection of input parameters and modelling approach can be achieved through calibration (often visual, McDougall, 2016); however this limits a quantitative elicitation of the effect model and input parameter choice have on the hazard prediction. Appropriate model performance evaluation would not only quantify this uncertainty, but also communicate the scale and spatial dependence of these effects, presenting opportunities for new hazard delineation methodologies. This paper presents a performance evaluation approach using a multi-scale fuzzy comparison technique that incorporates positional error tolerance to assess debris flow model performance. The 2012 debris flow from the Upper Te Maari crater, Tongariro volcanic centre (New Zealand) is used as a benchmark to test the effect of debris flow model choice on simulation accuracy. Given the necessity to assess model performance on the basis of its constructed purpose (Jakeman et al., 2006), each model is evaluated in terms of their ability to delineate hazardous debris-avalanche inundation zones, and demonstrate a new approach to define these zones with quantified uncertainty.

## 2 Case study: 2012 Upper Te Maari debris avalanche

The 6th of August, 2012 eruption in the Tongariro volcanic centre was a short (<60 s), but complex eruption sequence beginning with a slope failure on the outer, western flank of the Upper Te Maari Crater, followed by a series of (in order) East, West and vertically directed blasts that generated pyroclastic surges and ballistics covering an area greater than 6 km$^2$ (Jolly et al., 2014;Lube et al., 2014;Procter et al., 2014;Fournier and Jolly, 2014). The Te Maari debris avalanche emplacement, morphology and deposit characteristics were identified and summarised in Procter et al. (2014), here we summarise only the necessary details for this study.

The debris avalanche generated from the slope failure is presumed to have begun at 11:49:06 UTC, which is when an earthquake located at the avalanche head scarp is detected in the seismic record by Jolly et al. (2014). The cohesive, clay-rich debris avalanche was mostly confined by the Mangatipua Stream, travelling downslope to reach a run-out of approximately 2 km (Lube et al., 2014;Procter et al., 2014;Walsh et al., 2016). Volume of the debris avalanche was estimated between 6.83 and $7.74 \times 10^5$ m$^3$ in Procter et al. (2014), measured as the difference between a 10 m pre-event Digital Terrain Model (DTM) and post-event LiDAR derived DTM. The mud-sand matrix supported debris flow deposits were primarily emplaced in four lobes along Mangatipua Stream (see Figure 1). Coarse, poorly sorted, clasts ranging in size from pebbles to large boulders were also present, particularly at frontal lobes and lateral margins of the deposit, generally decreasing in quantity downstream. Between the lobes, steep channel sides limited deposition with 1 to 2 m of erosion into the soil substrate and thin (0.2 to 0.5 m) veneer deposits, demarcating maximum extents of the flow.

Geomorphic change associated with the debris flow was calculated through comparisons between the 10 m pre- and LiDAR derived post-event DTM. The pre-event DTM was created from contours generated from stereo-photogrammetry captured in 1975, with an accuracy of 90% within 10 m. The LiDAR post-event DTM was acquired on 8 – 9 November 2012 and has a 1-sigma accuracy of 0.25 m horizontally and 0.15 m vertically. While the accuracy of both terrain models are within acceptable limits, there are a number of issues in terrain model



interpolation, representation and acquisition time, which limits the comparison between terrain models and usage

in numerical modelling. Interpolation of contours to generate the pre-event DTM results in a smoothed representation of terrain with no sharp gradients which results in an inaccurate representation of steep features and narrow sections of Mangatipua stream (Procter et al., 2014). One major zone of misrepresentation is in the upper section of lobe 4, highlighted in Figure 1. In this region, elevation of the pre-event DTM increases to form a barrier to flow along Mangatipua stream which is not present in the field. The magnitude of this error is unknown,

but the height difference between pre- and post-event DTM's indicate it could be between 10 - 15 m. Additionally, the LiDAR survey used to generate the post-event DTM was taken 3 months after the debris flow, and almost 1 month after a breakout lahar (13 October 2012, Walsh et al., 2016) caused through damming of Mangatipua stream by the initial debris avalanche (Walsh et al., 2016;Procter et al., 2014). The breakout lahar, and possibly subsequent streamflow, eroded and entrained the 6[th] of August debris avalanche and ash deposits, redistributing

sediment further downstream and cutting a new stream into the deposit. Therefore, the post-event DTM does not represent the exact morphology of Mangatipua stream immediately after the debris flow.

Despite the previously mentioned uncertainties, the difference between the pre- and post-eruption DTM's form a useful data set for evaluating the accuracy of numerical models. Figure 1c shows the deposit outline (dotted black line) used in the accuracy assessment (see 'Model performance assessment' section) and points of deposit depth.

Deposit depth points are a subset of deposit depths in Procter et al. (2014) where the depth estimates are least affected by uncertainties in both terrain models. The points all have depths greater than 0.5 m (i.e. larger than LiDAR inaccuracies, avoiding veneer deposits) and are located where the pre-event terrain slope is moderate (<15 degrees) to avoid the effects of smoothing high gradient slopes in the pre-event DTM. An outline of the debris avalanche, shown as the black line in Figure 1, was created as the union of the debris avalanche outline in Procter

et al. (2014) and outline of the debris avalanche detected using image classifications from airborne hyperspectral survey in 2016. While this survey was undertaken 4 years after the eruption, thin veneer deposits appear to be detected and classified well (Kereszturi et al., 2018), improving the estimates of flow outline.

### 3 Debris avalanche simulation

#### 3.1 Numerical mass flow models

Numerical techniques to predict the motion of debris avalanches (and/or debris flows) commonly employ depth-averaging to simulate large scale geophysical flows (McDougall, 2016;Fischer et al., 2012), being favoured for their computational efficiency (in comparison to three-dimensional models), comparative scales, and level of detail to field measurements (Iverson and Ouyang, 2014). However, the physics of granular and granular-mixture flows is an area of active research and there are no universally accepted constitutive laws for debris flows

(McDougall, 2016). As a result, several models, varying in complexity from single-phase rheologies (e.g. Voellmy-Salm, Christen et al., 2010;O'Brien et al., 1993) to two-fluid (Pitman and Le, 2005) and multiphase approaches (Pudasaini, 2012;George and Iverson, 2014) have been used to simulate debris flows and avalanches (e.g. Iverson and George, 2016;Procter et al., 2010c;Mergili et al., 2017;Iverson et al., 2016;Sosio et al., 2007;Sosio et al., 2012;Sheridan et al., 2005).

For all models studied here, the depth averaged system of equations can be expressed in Cartesian coordinates as (Patra et al., 2005;Pudasaini, 2012):




$$\frac{\partial \mathbf{U}}{\partial t} + \frac{\partial \mathbf{F}}{\partial x} + \frac{\partial \mathbf{G}}{\partial y} = \mathbf{S} \tag{1}$$

Where **U** is the height and momentum vector, **F** and **G** are momentum fluxes in the $x$ and $y$ directions respectively, and **S** is the source term representing the net driving force (Patra et al., 2005;Patra et al., 2018). The three models

in our study utilise different assumptions and simplifications, and all four vectors (**U**, **F**, **G** and **S**) vary between models. The models studied here are the Pitman and Le (2005) two-fluid model and Voellmy-Salm rheology model (Salm, 1993;Fischer et al., 2012), both implemented in the Titan2D toolkit (Patra et al., 2005;Pitman et al., 2003), and the Pudasaini (2012) two phase model, implemented in the *Avaflow* software package (Mergili et al., 2017). The aim of this section is to summarise the features and key differences between each model that may

affect model comparison. Readers are referred to the source publications for complete model implementation details and justification of assumptions.

The Voellmy-Salm model (Salm, 1993) is a single phase rheological approach similar to shallow-water approaches, solved to find the unknown vector $\mathbf{U} = (h, hu, hv)^T$, where $h$ is the debris flow depth, and $u$, $v$ are the depth-averaged x- and y- direction velocities. The source term in this model assumes a combination of

coulomb-like basal friction, proportional to the coefficient $\mu$, and a velocity dependent turbulent friction, with coefficient $\xi$ (Christen et al., 2010). This combination of friction terms enables the simulation of both high and low velocity phases of the debris flow (Christen et al., 2010), but requires calibration of two coefficients ($\mu$, $\xi$) which may vary depending on topography and material properties (Fischer et al., 2012).

The Pitman and Le (2005) and Pudasaini (2012) approaches approximate the combined motion of granular

material and interstitial fluid, solving for the unknown momentum of both components (*fluids* in Pitman and Le (2005) and *phases* in Pudasaini (2012)). In the Pitman and Le (2005) approach, the vector $\mathbf{U} = (h, h\varphi, h\varphi u_s, h\varphi v_s, hu_f, hv_f,)^T$, where $\varphi$ is the solid volume fraction and subscripts $s$, $f$ indicate the solid and fluid components of velocity. The Pitman and Le (2005) model contains several features (comparted to the Voellmy-Salm model) that may affect the debris avalanche simulations:

i.   To account for the non-hydrostatic pressure distribution in granular materials (Scheidl et al., 2014), the earth pressure coefficient $K_{a/p}$ is used to relate the bed parallel ($\sigma_l$) to bed normal ($\sigma_n$) stresses. The active and passive earth pressures are calculated from the internal ($\phi_i$) and basal ($\phi_b$) frictions (Savage and Hutter, 1989;Pitman and Le, 2005;Iverson, 1997) depending on whether the flow is expanding (accelerating) or contracting (decelerating).

ii.  Solid (granular) and fluid (water) interaction is accounted for through a Darcy-like drag model and buoyancy effects, which alter energy dissipation within the flow.

This additional detail requires the specification of the volume fraction $\varphi$, and the granular materials internal ($\phi_i$) and basal ($\phi_b$) friction coefficients.

The Pudasaini (2012) model, in a different formulation to the Pitman and Le (2005), also accounts for buoyancy

effects, but also considers the effect of relative motion between fluid and granular phases through a 'virtual mass' term $\mathsf{C}$. This parameter, and the density ratio $\gamma = \frac{\rho_f}{\rho_s}$ are in the vector $\mathbf{U} = \left( h, h\varphi, (1-\varphi)h, h\varphi\left(u_s - \gamma\mathsf{C}(u_f - u_s)\right), h\varphi\left(v_s - \gamma\mathsf{C}(v_f - v_s)\right), (1-\varphi)h\left(u_f + \frac{\varphi}{1-\varphi}\mathsf{C}(u_f - u_s)\right), (1-\varphi)h\left(v_f + \frac{\varphi}{1-\varphi}\mathsf{C}(v_f - \right.\right.$




$v_s \big) \big) \big)^T$. While this formulation appears more complex than previous models, this approach still only contains six

unknown variables (i.e. as in Pitman and Le (2005) approach). The most notable differences in regards to debris

avalanche simulations are:

   i.    The virtual mass, $\mathsf{C}$, means the solid and fluid components of the debris flow are not assumed to be
         'interlocked' (relative velocity between phases of 0), and phase separation (such as at deposition) is
         accounted for, which is a key difference between the Pitman and Le (2005) and Pudasaini (2012) model.

   ii.   The source term contains a buoyancy-modified Coulomb term as in Pitman and Le (2005), and a
195       'generalised drag' term incorporating viscous drag effects. The generalised drag accounts for granular
         and fluid contributions to drag, the ratio of which depends on the interpolation parameter $\mathcal{P}$ with an
         exponent, $\mathcal{I}$, controlling whether the drag term is linear or quadratic (i.e. similar to Voellmy-Salm
         models). At $\mathcal{P}=0$ and $\mathcal{I}=1$, the Pitman and Le (2005) drag model is recovered.

The generalised drag term and virtual mass coefficient extends the applicability of the model for all types of

granular-fluid flows, including at extremes (i.e. high or low solid fraction flows), but requires the specification of

14 parameters. While many of these parameters can be specified from field/material properties (see e.g. Mergili

et al., 2017), some values (e.g. $\mathcal{I}$, $\mathcal{P}$, $\mathsf{C}$) require calibration.

From an application point of view (i.e. neglecting differences in numerical solution techniques), these models

vary in their level of description ('completeness') of debris flow physics. It is important to identify that even well

observed and quantified debris flows, such as the one studied here, may have considerable uncertainty in material

properties, which can expand the unknown (i.e. to be calibrated) parameter space. Therefore, while some models

may offer more complete descriptions of debris flow physics, there may not be a commensurate improvement in

prediction compared to less complex models when uncertainty in material properties is considered. Therefore, this

study analyses the performance of all three models to elicit, under locational uncertainty, the relative

improvements and trade-offs in accuracy considering calibration and parameterisation needs.

### 3.2 Initial and boundary conditions

The DTM, location and height of debris avalanche source material are common inputs to all debris avalanche

simulators in this study. The DTM input was defined from the 10 m pre-event terrain model, modified to: (a)

remove the previously discussed misrepresentation of elevation (see Figure 1) along Mangatipua stream, and (b)

remove the debris avalanche source from the terrain model. The spurious elevation was modified by adjusting

elevations in this region to equal the post-event LiDAR survey elevations. Terrain model elevations in the source

area were also adjusted to account for debris avalanche material to be simulated, using source depths from Procter

et al. (2014), which were used as the input for the initial pile of debris avalanche material.

The parameters used in each debris avalanche model simulation are shown in Table 1. Selection of these values

were derived from previous examples of debris avalanche simulations in literature and the authors' experience

(e.g. Mergili et al., 2017;Sosio et al., 2012;Mead and Magill, 2017), and visual comparisons to flow and deposit

properties (i.e. similar to visual calibration in McDougall, 2016). Since the aim of this study is to demonstrate

performance evaluation to delineate hazard zones *a-priori*, we chose not to undertake further calibration of

rheological parameters (such as in Charbonnier et al., 2017). This best represents typical conditions where the

only data available are flow and deposit outlines.




### 3.3 Simulation results

Snapshots of the simulated debris flow depth are shown in Figure 2 for each model. Simulated debris flow behaviour is generally similar for all three modelling approaches, being an acceleration of material from rest at the debris avalanche source to the upper reaches of Mangatipua stream (0 – 40 seconds), where debris flow

material ponded (~40 – 100 seconds, not shown) and gradually travelled downstream to its rest (150 – 300 seconds). This is in agreement with field and LiDAR based interpretations of the event (Procter et al., 2014), and best-fit volume fraction (0.8 – 0.85) parameters of the Pitman and Le (2005) and Pudasaini (2012) support the hypothesis of an unsaturated debris flow. Visual, qualitative comparisons of flow and deposit outlines also appear to match reasonably well to observations. While the Voellmy-Salm simulation shows less ponding in upper

Mangatipua stream and has a more defined (i.e. steeper) distal deposit compared to the other simulations, other differences between simulations appear minor.

Figure 3 shows the simulated deposit depth (at 300 seconds) for each model. Black lines indicate the observed deposit and source outlines from Procter et al. (2014). Accurate prediction of deposition is difficult in these depth-averaged approaches as none explicitly consider stopping of material (Mergili et al., 2017). The predicted deposits

for all simulations appear (qualitatively) to have similar levels of accuracy as other depth-averaged debris flow case studies (e.g. Rickenmann et al., 2006;Iverson and George, 2016;Charbonnier et al., 2017). The most notable difference in simulated deposits is between Voellmy-Salm (i.e. a rheological approach, Figure 3a) and the two-fluid approaches (Figure 3b,c). The Voellmy-Salm simulated deposit is mostly confined to the Mangatipua Stream, whereas the two-fluid approaches show more spreading of the distal deposit and shallow flow in areas of

super elevation.

### 4 Model performance assessment

The previous qualitative comparisons between field observations and simulation results can provide some credibility to flow predictions; however, modern, robust hazard assessments require quantitative evaluations of model performance to understand the level of uncertainty in model predictions. Previous assessments of flow

simulation accuracy (Mergili et al., 2017;Charbonnier et al., 2017), use various ratios of data points classified as either true negative (TN), false negative (FN), true positive (TP) or false positive (FP) to quantify aspects of accuracy as a single value (between 0.0 and 1.0). Map cell classification is achieved through a pairwise comparison of simulation results and the observed flow outline (solid line in Figure 1c), as illustrated in Figure 4. There is a wide range of ratios (see e.g. Sing et al., 2005) to quantify model accuracy, and there is usually no

single 'best' metric (Charbonnier et al., 2017). Rather, several metrics are usually analysed together to achieve a comprehensive understanding of performance. However, as demonstrated in Figure 4, the proportion of TN values is dependent on size of the model domain, and can have significantly higher counts than other values. Therefore, for model evaluation purposes, most metrics that consider TN values in the denominator or numerator are unsuitable. Here, as with previous approaches (Charbonnier et al., 2017;Mergili et al., 2017), we calculate positive

predictive value (PPV), sensitivity and critical success index (CSI, Formetta et al., 2016; also called the Jaccard similarity coefficient) performance metrics for each flow model.

Figure 5 shows the calculated performance metrics for each flow simulation at depth cut-off values up to 10 m. The depth cut-off value is the threshold to convert simulated flow depths into a binary (i.e. inundated/not




inundated) classification, and performance curves were calculated using the ROCR package (Sing et al., 2005) in
the R statistical language. The PPV, shown in Figure 5a, is the proportion of correctly simulated (TP) areas within
the simulated inundation footprint, calculated as $PPV = \frac{TP}{TP+FP}$. As FP values are in the denominator, this measure
penalises over-prediction of debris flow extents. This metric increases with depth cut-off values, indicating it is
less likely that simulations are incorrect in areas where deep flow is predicted. Sensitivity (Figure 5b), the
proportion of correctly simulated (TP) areas within the observed inundation footprint ($Sens = \frac{TP}{TP+FN}$), penalises
under-prediction of debris flow extents and shows the opposite trend. The CSI (Figure 5c) penalises both under-
and over- prediction flow extents, calculated as the proportion of correctly simulated (TP) areas within a combined
simulation and observed inundation footprint ($CSI = \frac{TP}{TP+FP+FN}$). Values of this metric are much lower than the
sensitivity or PPV as both under- and over-prediction are penalised. In effect, this metric identifies the proportion
of the simulated inundation footprint that is correctly simulated. In effect, this means values of CSI less than 0.5
indicate the simulation is more 'incorrect' than 'correct' (more FP and FN than TP).

In Table 2, we report performance metrics at a depth cut-off where CSI indicates the simulations are more 'correct'
than 'incorrect' (0.5 m). The accuracy metrics reported here are comparable to similar flow simulation studies
where accuracy metrics are explicitly reported (e.g. Charbonnier et al., 2017;Mergili et al., 2017). However, as
previously discussed, single-value metrics are useful for model comparison, but can conflate and disguise sources
and the distribution of error. In particular, the spatial distribution of error is not random; rather, it is related to
topography (e.g. degree of confinement to the channel) and distance from source. To demonstrate this effect, the
maximum flow depth for each cell (10 m resolution) was mapped to its corresponding PPV (i.e. from Figure 5)
and is shown in Figure 6. At the centre of the debris flow, where flow depth is high, PPV's are generally higher
and the largest area of low PPV's for all simulations are most distal from the source (where flow depth is low).
This indicates trends of decreasing PPV away from the source and topography affecting PPV, with areas of low
topographic slope having lower PPV's. The correlation of flow depth and PPV from topographic and distance
from source effects result in a degree of spatial autocorrelation in the performance metrics that are un-reported in
global PPV measures.

### 4.1 Locational tolerance in performance assessment

The uncertainties in terrain data, initial conditions, model assumptions and potential observation errors suggest
that precise quantitative and locational agreement between simulated and observed debris flows are unlikely.
Some level of error tolerance is therefore necessary in comparisons, and would produce performance metric values
more aligned with qualitative (i.e. human) assessments (Wealands et al., 2005). The spatial autocorrelation of
performance metrics, shown in Figure 6, indicate that locational tolerance may be accounted for by considering
the cell neighbourhood in model comparisons.

To incorporate the influence of neighbouring cells in defining inundation footprints for model comparison, a fuzzy
set approach is used. Our approach, based largely on Hagen (2003), expands classification boundaries according
to a distance decay function. The classification of each cell is converted into a 'fuzzy' membership vector,
calculated as $max(c_i \cdot W)$, where $c_i$ is the 'crisp' category membership vector (e.g. 1,0 for a map with
inundated/not inundated categories) and $W$ is the distance decay function. Here, we use a 2-dimensional Gaussian-
like weighting function:



$$W_i = e^{\frac{-\left(i - \frac{\lambda - 1}{2}\right)^2}{2\sigma^2}} \tag{2}$$

where $W_i$ is weighting of cell $i$, $\lambda$ is the neighbourhood width/length (in cells), $i = 0 \dots \lambda - 1$, and the standard
deviation is defined as a function of the neighbourhood size:

$$\sigma = 0.3\left(\frac{\lambda - 1}{2} - 1\right) + 0.8. \tag{3}$$

This technique creates a 'fuzzy' quantity (between 0 and 1) that indicates the proximity of a cell to similar-valued
cells.

Figure 7 shows the performance metrics fuzzified through equations 2 and 3. The target performance metric
(sensitivity, Figure 7a) is improved by increasing the length scale ($\lambda$) to account for spatial autocorrelation. The
increase in sensitivity is associated with a commensurate decrease in the opposing performance value (PPV),
shown in Figure 7b. The greatest change in performance occurs between length scales of 10 m (i.e. model scale,
no fuzzification) and 30 m (i.e. 3 model cells), and the rate of change diminishes with length scale, with only
marginal improvements in performance beyond 70 m (7 model cells). This indicates the approximate scale of
positional error in the model, demonstrating the length scale parameter could be an alternative metric to quantify
performance and evaluate models to their desired purpose. For example, a 'best' model could be chosen from the
model with the smallest length scale that exceeds a desired performance threshold, or through optimising the
trade-off between sensitivity and PPV beyond a given threshold.

## 5 Discussion

### 5.1 Model suitability, calibration and performance

In this study, we restricted our use of post-event calibration to a visual calibration within a limited bound of
reasonable values. Despite this restriction, the potential improvement in model performance appears marginal
when compared to more extensive calibration procedures such as parameter sweeping (e.g. a CSI of ~0.6 reported
in Mergili et al., 2017) or rheological calibration (e.g. CSI of ~0.5 reported in Charbonnier et al., 2017). These
differences are on a similar scale to the performance differences between models in our study. Potentially, this
indicates source and input (e.g. terrain) uncertainties are a greater limitation on performance than uncertainties
introduced by model and parameter choice. The spatial autocorrelation in performance values (Figure 6) also
supports this assertion. Areas of deeper flow (centre of channels) are likely to be less affected by terrain
uncertainties than those with shallower flows (at edges of channels). In contrast, performance of the entire
simulation domain would be affected if model choice or poorly calibrated parameters were the primary source of
error. This is not evident in simulations presented here, and the improvement in PPV when accounting for
autocorrelation further suggests terrain error, not model error, is dominant in real-scale mass flow simulations.

### 5.2 Implications for hazard zonation

An advantage of the previously described fuzzy performance evaluation approach is the identification of
simulation uncertainty at various length scales. This presents and avenue to generate hazard zonation's and maps
that incorporate areal uncertainty. For example, Figure 8 shows debris avalanche hazard outlines generated at



model scale (10 m) and a length scale of 70 m (see caption for details on delineation of the zones). The false negative rate (i.e. 1 - model sensitivity) decreases from 0.07 to 0.01 between model and fuzzified estimates, a crucial reduction from a life safety perspective. The technique also identifies trade-offs in minimising the false negative rate, such as an increase in area and decrease in positive predictive value from 0.52 to 0.46.

The hazard zonation approach (and fuzzification technique in general) demonstrated here can address key issues in the generation of volcanic hazard zonation's by:

- quantifying and communicating model-based uncertainty, including the trade-off between hazard zone area, positive predictive value and sensitivity, and

- identifying appropriate scales at which to communicate simulated hazard data. For example, if the chosen length scale is 70 m, an appropriate map scale is approximately 1:70,000 (Tobler, 1987).

This method can also reduce the complexity of estimating uncertainty and making decisions using simulated data to a problem that only requires judicious choice of a length scale value. However, the lack of a currently defined (mathematical) basis to parameterise length scale means that a careful consideration of hazard exposure and the acceptability of risk to exposed elements is required. This may differ between users; we therefore believe the hazard zonation process is best suited for use by informed stakeholders as a decision support tool, rather than a process to generate publically disseminated hazard zones.

## 6 Conclusion

The accuracy of three depth-averaged numerical models were assessed using the 2012 Te Maari debris avalanche as a benchmark. Results of the simulations show a similar qualitative accuracy of all three models to other published studies. Quantitative performance metrics of inundation area show high model sensitivity (i.e. a low proportion of false positives) with moderate values of positive predictive values and the critical success index, which are similar in scale to other published performance assessment studies.

Our investigation also demonstrates the positional dependence of model performance, specifically the positive predictive value, where model performance (i.e. accuracy) is highest in areas of deep flow or where topography is steep with confined channels. Using this observation, a fuzzy set approach is used to incorporate locational tolerance (the covariance of location and positive predictive value) into the simulation performance assessment. We found increasing the length scale ($\lambda$) of the correlation function can increase performance metrics, for a commensurate decrease in the opposing performance metric (e.g. an increase in sensitivity leads to a decrease in positive predictive value) and decrease in resolution.

The identification of positional uncertainty in hazard simulations has positive implications for hazard zonation and mapping. The process demonstrated here can improve desired performance metrics (e.g. sensitivity), account for uncertainty (by increasing hazard zone area) and identify trade-offs to opposing metrics (e.g. positive predictive value). This can be a valuable tool for informed stakeholders with well-quantified exposures and risk tolerances. The process is, however, less suited to publicly disseminated hazard information due to the lack of a mathematically optimum solution for length scale. An optimum solution may be identifiable, and progress in hazard zonation methodologies would benefit from deeper investigation of the trade-offs between area, length scale and model performance to fully leverage benefits of the fuzzy performance evaluation approach.



**Acknowledgements**

This research was supported by Natural Hazards Research Platform (NHRP) ("Too big to fail? A multi-disciplinary approach to predict collapse and debris flow hazards from Mt Ruapehu") and the Resilience to Natures Challenges National Science Challenge "Volcanism" science programme. The authors wish to acknowledge the use of New Zealand eScience Infrastructure (NeSI) high performance computing facilities as part of this research. New Zealand's national facilities are provided by NeSI and funded jointly by NeSI's

collaborator institutions and through the Ministry of Business, Innovation & Employment's Research Infrastructure programme. Code to generate fuzzy membership vectors is available from the Zenodo data repository with doi: 10.5281/zenodo.2797984

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



**Tables**

| Voellmy-Salm | | Pitman and Le (2005) | | Pudasaini (2012) | |
|---|---|---|---|---|---|
| **Parameter** | **Value** | **Parameter** | **Value** | **Parameter** | **Value** |
| Basal friction coefficient, $\mu$ | 0.15 | Basal friction angle, $\phi_b$ | 31° | Basal friction angle $\phi_b$ | 21° |
| Turbulent friction coefficient, $\xi$ | 1091 | Internal friction angle, $\phi_i$ | 36° | Internal friction angle, $\phi_i$ | 36° |
| | | Solid volume fraction, $\alpha$ | 0.80 | Solid volume fraction, $\alpha$ | 0.83 |
| | | | | Virtual mass, $C$ | 0.5 |
| | | | | Solid material density, $\rho_s$ | 2,500 |
| | | | | Fluid material density, $\rho_f$ | 1,000 |
| | | | | Fluid-solid drag contributions, $\mathcal{P}$ | 0.5 |
| | | | | Fluid-solid drag exponent, $\mathcal{J}$ | *linear* |
| | | | | All other parameters as in Mergili et al. (2017), Table 2 | |

**Table 1. Debris flow simulation parameter settings for all models.**

| Simulation | Positive predictive value (PPV) | Sensitivity | Critical success index (CSI) |
|---|---|---|---|
| Voellmy-Salm | 0.62 | 0.93 | 0.59 |
| Pitman and Le (2005) | 0.53 | 0.93 | 0.51 |
| Pudasaini (2012) | 0.51 | 0.98 | 0.51 |

**Table 2. Pixel-pair performance assessment results for all flow simulations at a depth cut-off of 0.5 m.**





**Figures**

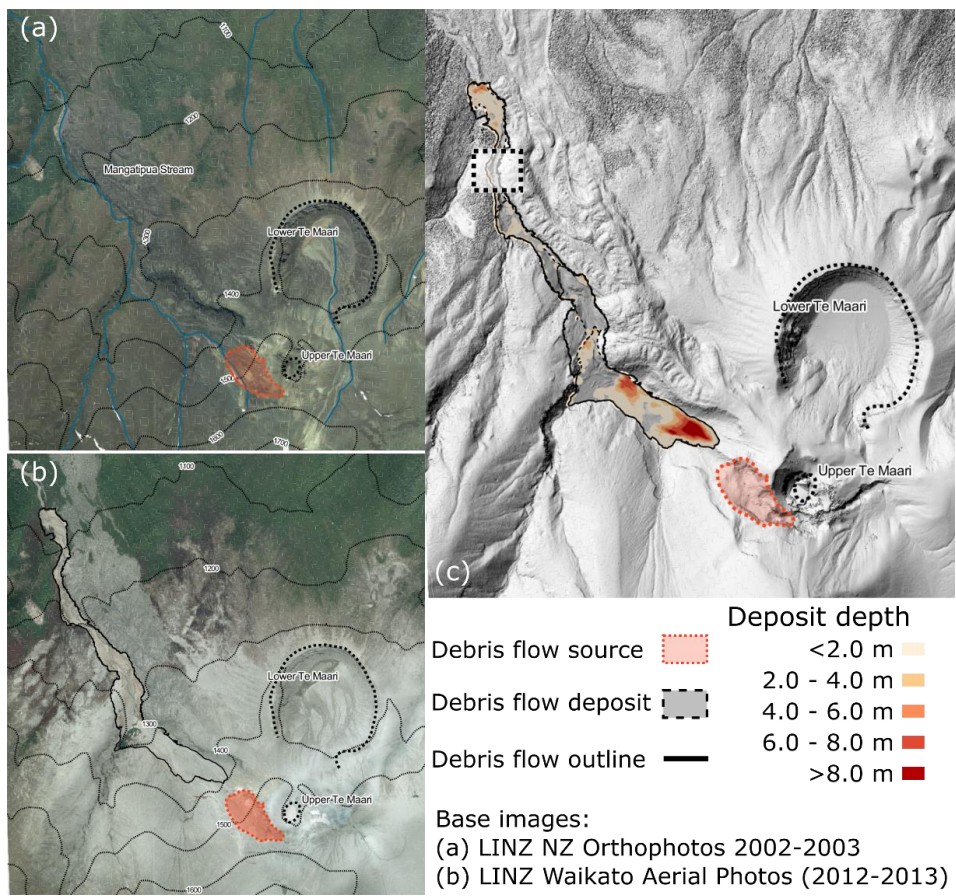


**Figure 1. Te Maari debris avalanche case study region (a) pre-eruption, (b) post-eruption, and (c) debris avalanche deposit depth and outline. Dashed rectangle in (c) shows area of spurious elevations from source Digital Terrain Model.**

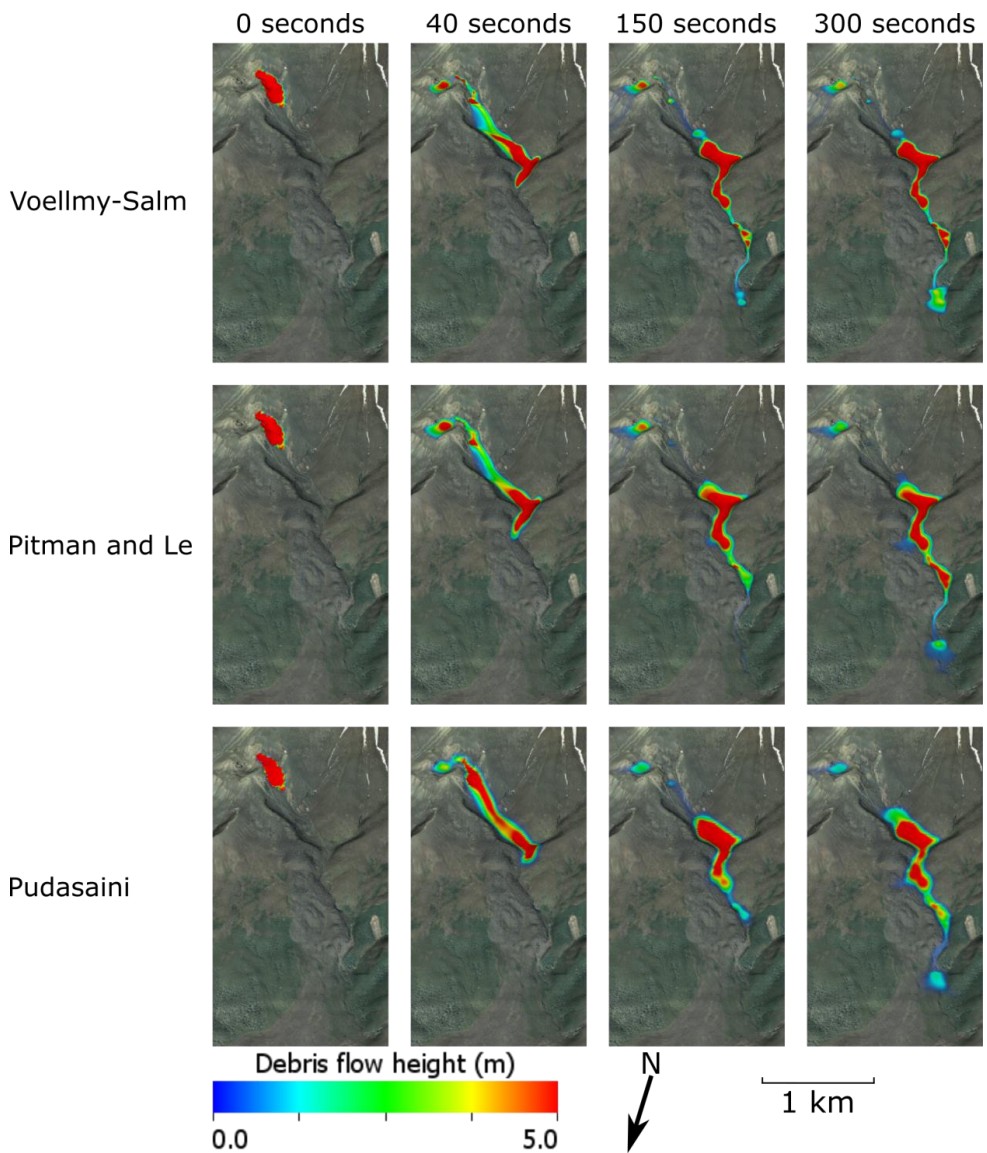

**Figure 2. Snapshots of simulated debris flow height for each flow model at 0, 40, 150 and 300 seconds after initiation. Aerial basemap sourced from LINZ Waikato Aerial Photos (2012-2013).**


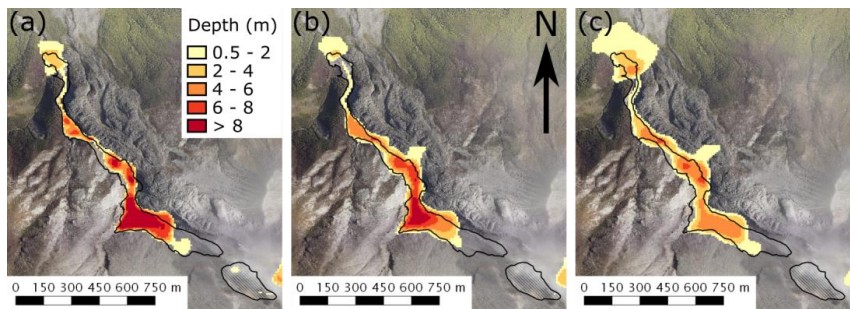

**Figure 3. Simulated deposit depth for (a) Voellmy-Salm, (b) Pitman and Le (2005), and (c) Pudasaini (2012) models compared with the observed deposit and source outline (black). Aerial basemap sourced from LINZ Waikato Aerial Photos (2012-2013).**




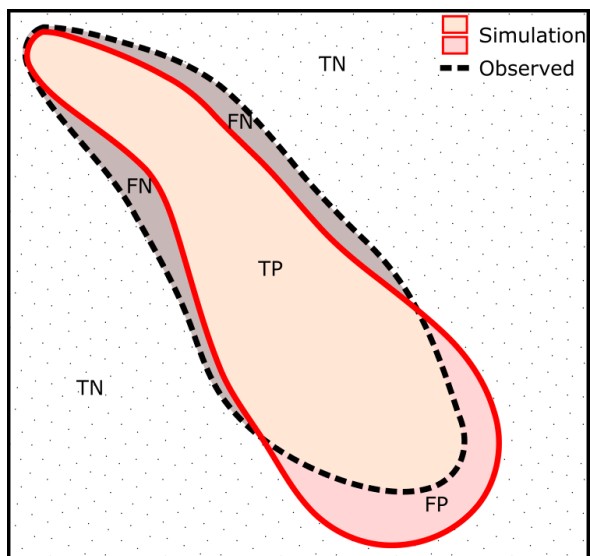

**Figure 4. Illustration of confusion matrix classification for simulation performance assessment. Dashed black outline represents the observed flow outline; solid red outline represents the simulated flow outline. Areas outside of both simulated and observed flow outlines are classed as True Negatives (TN, dotted region); areas outside simulated outline but inside observed outline are classed as False Negatives (FN); areas inside both simulated and observed outline are classed as True Positives (TP); areas inside simulated outline but outside observed outline are classed as False Positives (FP).**


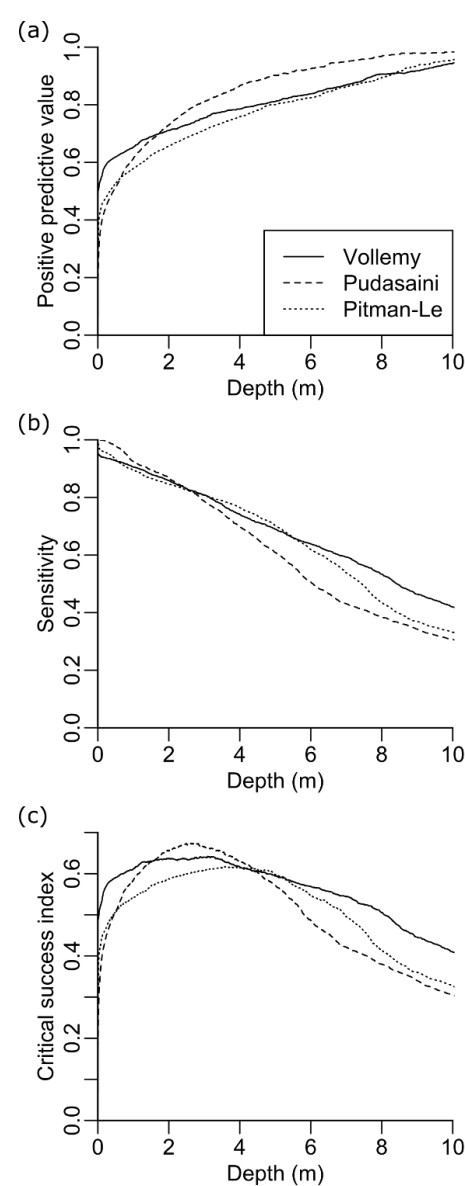


**Figure 5. Flow outline performance with depth for Voellmy-Salm (solid line), Pudasaini (2012) (dashed line) and Pitman and Le (2005) (dotted line) flow models. Performance metrics are: (a) positive predictive value (PPV), (b) Sensitivity, and (c) Critical success index (CSI).**

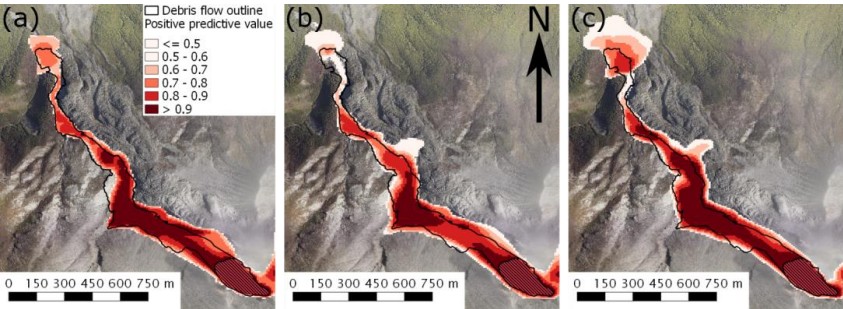

**Figure 6. Positive predictive values for (a) Voellmy-Salm, (b) Pitman and Le (2005), and (c) Pudasaini (2012)**
**simulations. The observed deposit and source are outlined in black. Aerial basemap sourced from LINZ Waikato Aerial**
**Photos (2012-2013).**


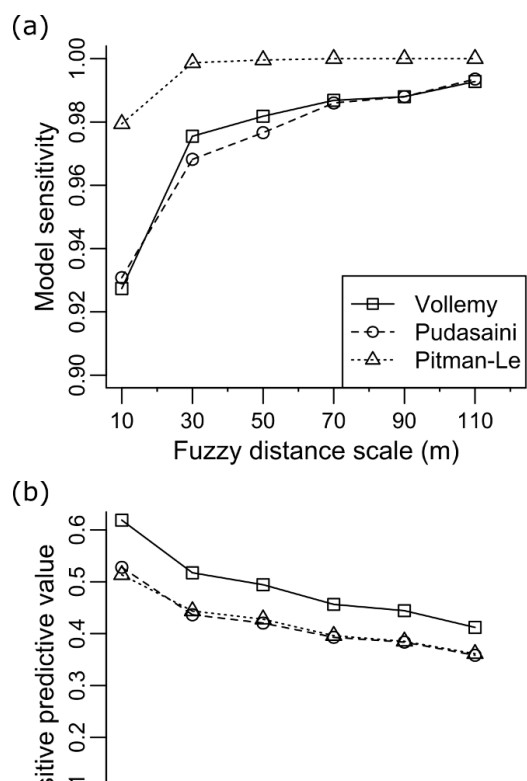

**Figure 7. Fuzzy performance metrics at 3 cell (30 m), 5 cell (50 m), 7 cell (70 m), 9 cell (90 m) and 11 cell (110 m) length scales ($\lambda$) for (a) model sensitivity and (b) positive predictive value for Voellmy-Salm (solid line), Pudasaini (2012) (dashed line) and Pitman and Le (2005) (dotted line) flow models. In these graphs, a cell is considered inundated if its fuzzy quantity is greater than 0.25.**


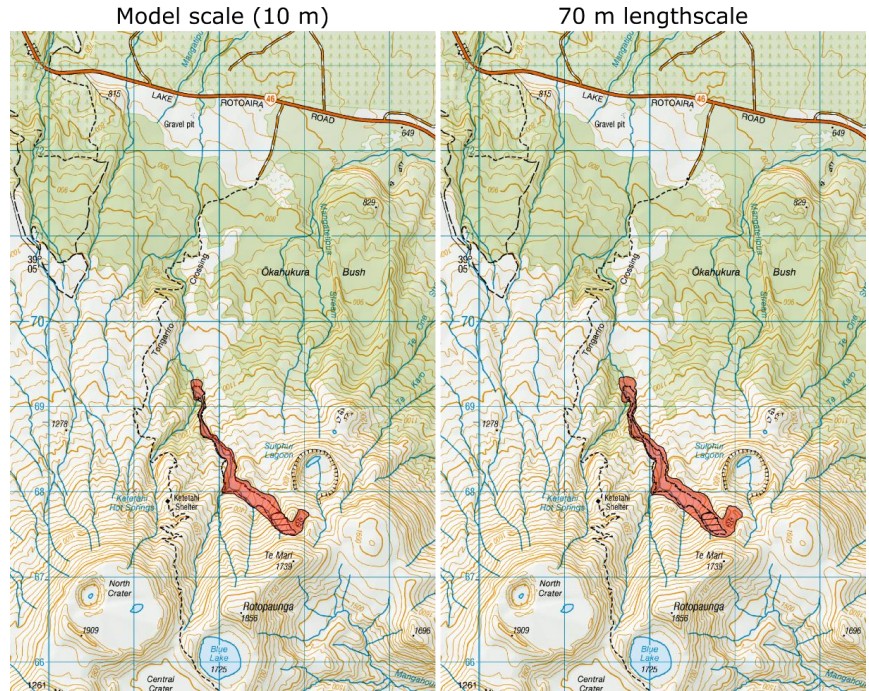

**Figure 8. Hazard zones generated from simulations at (left) 10 m model scale, and (right) using fuzzy length scale of 70 m. Hazard outline for model scale is generated where flow heights exceed 0.5 m. Hazard outline for 70 m scale is generated where fuzzy quantity exceeds 0.25. Hazard zones are overlain on New Zealand Topo50 map from Land Information New Zealand (LINZ), blue gridlines are 1 km apart, oriented North-South and East-West.**
