# Peer review of "Quantifying location error to define uncertainty in volcanic mass flow hazard simulations"

_Natural Hazards and Earth System Sciences, 2021_

## Author Comment (AC3)

**Figures**

[Figure]

**Figure 1. Te Maari debris avalanche case study region (a) pre-eruption, (b) post-eruption, and (c) debris avalanche deposit depth and outline. Blue rectangle in (c) shows area of spurious elevations from source Digital Terrain Model.**

[Figure]

Figure 2. Snapshots of simulated debris flow height for each flow model at 0, 40, 150 and 300 seconds after initiation. Aerial basemap sourced from LINZ Waikato Aerial Photos (2012-2013).

[Figure]

**Figure 3. Simulated deposit depth for (a) Voellmy-Salm, (b) Pitman and Le (2005), and (c) Pudasaini (2012) models compared with the observed deposit and source outline (black). Aerial basemap sourced from LINZ Waikato Aerial Photos (2012-2013).**

[Figure]

**Figure 4. Illustration of confusion matrix classification for simulation performance assessment. Dashed black outline represents the observed flow outline; solid red outline represents the simulated flow outline. Areas outside of both simulated and observed flow outlines are classed as True Negatives (TN, dotted region); areas outside simulated outline but inside observed outline are classed as False Negatives (FN); areas inside both simulated and observed outline are classed as True Positives (TP); areas inside simulated outline but outside observed outline are classed as False Positives (FP).**

15

[Figure]

**Figure 5. Flow outline performance with depth for Voellmy-Salm (solid line), Pudasaini (2012) (dashed line) and Pitman and Le (2005) (dotted line) flow models. Performance metrics are: (a) positive predictive value (PPV), (b) Sensitivity, and (c) Critical success index (CSI).**

20

[Figure]

**Figure 6. Positive predictive values for (a) Voellmy-Salm, (b) Pitman and Le (2005), and (c) Pudasaini (2012) simulations. The observed flow and source are outlined in black. Aerial basemap sourced from LINZ Waikato Aerial Photos (2012-2013).**

[Figure]

**Figure 7. Fuzzy performance metrics at 3 cell (30 m), 5 cell (50 m), 7 cell (70 m), 9 cell (90 m) and 11 cell (110 m) length scales ($\lambda$) and fuzzy quantities of 0.1 (blue), 0.25 (black) and 0.5 (red) for (a) model sensitivity and (b) positive predictive value for Voellmy-Salm (solid line), Pudasaini (2012) (dashed line) and Pitman and Le (2005) (dotted line) flow models.**

30

[Figure]

**Figure 8. Hazard zones generated from simulations at (left) 10 m model scale, and (right) using fuzzy length scale of 70 m. Hazard outline for model scale is generated where flow heights exceed 0.5 m. Hazard outline for 70 m scale is generated where fuzzy quantity exceeds 0.25. Hazard zones are overlain on New Zealand Topo50 map from Land Information New Zealand (LINZ), blue gridlines are 1 km apart, oriented North-South and East-West.**

---

## Author Response (AR1)

**Reviewer 1:**

This is an excellent paper which clearly outlines a new approach to quantifying uncertainty in debris flow simulation. The paper is well written and the approach is easy to follow. The outlined approach has the potential to be applied to other volcanic and non-volcanic hazards and has positive benefits for hazard mapping and hazard zone creation. The paper concludes by raising questions around acceptable risk, a potential area for future investigation.

The approach applies a 'fuzzification' to simulated flow boundaries. As the length scale of the fuzziness increases, the PPV (proportion of true positives within the simulated footprint) decreases and the hazard area is increased, thereby accounting for uncertainty. This means that as the length is increased, the model results are more conservative from a life-safety point of view.

I see very few problems with the manuscript and believe that this is an excellent contribution to the literature. Limitations are clearly elucidated.

Two small points:

Line 262 and beyond doesn't read well – referring to values up to a 10 m flow depth – may be better explained by "The depth cut-off values are the thresholds to convert simulated flow depths…"

**Response**

The repetition of depth is confusing here and is better defined as an 'inundation threshold'. We have changed the text to use 'inundation threshold', the value of which defines the inundated/not inundated classification.

Line 335 = "and" should be "an"

**Response**

Changed to 'an'

**Reviewer 2 (Jessica Ball)**

This paper is a valuable contribution to debrisflow modeling on volcanoes, and takes important steps toward crafting a method for the quantification of uncertainty in debris flow modeling results. The results of this study will help make model results more robust and contribute to better hazard maps, and it is nearly ready for publication; addressing the comments below will strengthen the writing and make it easier for other practitioners to make use of the methods described. In addition, some minor modifications to the figures will make them easier to read and interpret. Minor typos are noted at the end of the more substantive comments.

**Comments**

Line 104 (Figure 1): This figure contains multiple sets of black dotted lines (crater outlines, debris flow deposit, contours, and spurious elevation zone), which is confusing to interpret. The craters are apparent and only need labeling (and perhaps arrows); the contours, deposit, and spurious zone should all be different designs and/or colors; and all of the labels need to be larger in order to be readable.

**Response:**

We have made changes to Figure 1, caption and text based on these suggestions to improve interpretation.

Line 227 (Figure 2): Using a single color gradient rather than a rainbow here would make this figure much easier to interpret. Rainbows are subject to misinterpretation (see http://iis.seas.harvard.edu/papers/2011/borkin11-infoviz.pdf) and it's not easy to see the subtle variations in these simulations based on a rainbow colorscale.

**Response:**

We have changed Figure 2 to use a better color scheme for the sequential data, based on colorbrewer. This scheme is also consistent across all depth measures (Figure 3, categorical) as suggested in following comments.

**Comment:**

Line 237 (Figure 3): There's no depth comparison going on here, only a comparison of inundation limits. Can you show the difference between the simulated and actual deposit depths as a gradient layer?

**Response:**

Around line 240, we have modified text to highlight the comparison is between deposit outlines ( "The predicted deposit **outlines** for all simulations …"). We are unable to reliably extract a difference between simulated and actual deposit depths due to the differences in post-event LiDAR which does not represent the immediate post-event morphology (see lines 109-127). The modelling approaches also do not explicitly consider stopping of material (see line 239), which would make quantitative comparisons suspect. This limits the analysis to a qualitative assessment of inundation limits.

**Comment:**

Additionally, this is a different gradient than in the previous image. It's better (see comment about rainbows) but it would be nice to see some consistency in colors depending on the variable being shown (depth, PPV, etc.) Also, the outlined/white dotted zone at the bottom right needs a label, assuming it is the source zone for the simulations.

**Response**

Colorscales are now similar for heights (Fig. 2 & 3; note one is continuous vs. categorised) and PPV is a separate color gradient (Fig. 6).

**Comment:**

Line 276 (Table 2): Why do the basal & internal friction angles and the solid volume fraction parameters vary between these two models? In a first-order comparison of their ability to accurately represent a flow deposit, wouldn't you want to make sure that all the shared parameters are consistent between the different models? If not, the reason for these choices needs to be documented in the methods section.

**Response:**

(we presume you mean ~Line 219/Table 1) The models themselves vary in completeness of the model physics or take a rheological approach, which result in a difference between shared parameters. The large difference between Voellmy and two-phase model basal friction is due to the application of basal friction to the solid fraction only. The calibration approach (as in McDougall 2016) may also result in small differences between the models, but the larger part of the difference is due to the different models applied. This has been clarified in text (*"Best-fit values for similar parameters (basal friction and solid volume fraction) in Table 1 vary as a result of the differing drag contributions, parameter sensitivity, rheological and constitutive models and the calibration approach. For example, the two-phase models only apply basal friction to the solid volume fraction, whereas the (single phase) Voellmy-Salm approach considers a bulk basal friction of the fluid-solid mixture, and additional viscous stresses in the Pudasaini (2012) model appeared to reduce the sensitivity and value of basal friction."*).

**Comment:**

Line 283 (Figure 6): The debris flow outline is difficult to see here. Perhaps a colorscale that's less saturated on the high end, or a diverging one?

**Response:**

We have applied a less-saturated version of the PPV colorscale

**Comment:**

Line 321-322: What are 'reasonable' values? What is reasonable to one practitioner may not be for another, or in a different setting.

**Response:**

We have revised the text to clarify the values are within the range of those identified in previous literature.

**Comment:**

Line 336 (Figure 8): How did you choose 0.25 as your fuzzy quantity limit? What makes this value important from a hazard zonation standpoint? Additionally, the figure appears to show hazard zones overlain on the debris flow deposit outline, but there's no label distinguishing the two outlines or pointing out the source area (assuming it is the cross-hatched zone at the upper end of the flow).

**Response:**

The fuzzy quantity has a similar effect to length scale, but has more variability at small length scales due to the discrete approximation of a gaussian. We chose to set the quantity to 0.25 as the smallest value possible in a 3-cell weighting kernel. This needed to be clarified, so we have modified Figure 7 to show fuzzy quantities of 0.1, 0.25 and 0.5, and added a short explanation to the end of section 4.1.

Figure 8 has been modified to add labels.

**Comment:**

Line 337: Is the 'length scale' being referred to here the length scale of the correlation function? You may wish to specify so as to avoid confusion.

**Response:**

Yes, length scale of the correlation function (equation 2). Modified in text.

**Comment:**

Line 345: When you say "map scale", do you mean the resolution of the model results, or the DTM? Can you explain how you derive the map scale, since the reference you include is old and not available online? It may seem simplistic to an experienced practitioner, but going into more detail will mean that other researchers will be able to more easily apply your methods.

**Response:**

"Map scale" here refers to the scale of the hazard map (i.e. including all features it may be important to include in a hazard map for communication). We have modified wording to suggest this. We have also added a short explanation for how the map scale denominator is obtained (by multiplying length scale by 1000) from the Tobler method.

**Comment:**

Line 350-352: I find this wording confusing. Why else would you put the effort into evaluating the output of models if they aren't going to be used to generate hazard zones in the long run? Do you mean instead, "this method is useful for quantifying the uncertainty of modeling output, which can be assist in generating hazard zonations according to specific risk

tolerances, but does not produce raw data appropriate for public products"? Lines 367-370 are a less confusing way of stating this, and would do better in place of the current text.

**Response:**

This (lack of a mathematical basis to parameterise length scale) is a limitation of our method as currently defined, we think this limitation should be communicated largely as written. To reduce wording confusion, we have made minor modifications to text on these lines to emphasise it is a *current* limitation that could be alleviated and the method cannot be used as an *automated* process to generate hazard zones for uninformed stakeholders (i.e. 'public').

**Comment:**

Line 363: Worded confusingly - break this up into a couple of clear statements to drive the point home.

**Response:**

Broken up into a clear statement and example.

*Text errors have been addressed:*

Line 32: 'models' should be 'model's'

Line 335 (and others): 'hazard zonation's' should be 'hazard zonations'

**Reviewer 3 (David Jessop)**

This is a review of "Quantifying location error to define uncertainty in volcanic mass flow hazard simulations" by Mead et al. This manuscript aims to quantify "location error", that is the difference between observed and simulated deposit locations. A lack of model uncertainty quantification, i.e. location error, along with "model complexity", i.e. uncertainty over the best-adapted physical model or correct values of model parameters are cited as reasons that mass flow simulations are not more widespread in volcanic hazard assessment. Thus efforts to quantify and/or constrain these errors are to be applauded. This work is such an efford. It takes a novel fuzzy logic approach to the quantification. Several common rheological models for debris flow modelling are assessed within this framework. Data from the 2012 debris avalanche of the Upper Te Maari crater, Tongariro volcano, NZ is used as a benchmark in this study. I find that the study is generally interesting and novel, though some points require clarification.

**Comments:**

L038-041: The implication is that scaled, experimental models are (over)simplified and hence unsuitable as benchmarks, as opposed to real-world flows. I think the truth is somewhere inbetween: real-world flows (or more realistically their deposits) are often subject to erosion, slumping or alteration before being surveyed.

**Response:**

We have modified the wording on these lines to reduce this inference by readers ("Experimental facilities and studies … can provide detailed observations of mass flow processes to validate, develop and benchmark numerical models") and clarified 'real world' mass flows are useful for assessing model accuracy at a subsystem level (e.g. as suggested by Esposti Ongaro et al., 2020), rather than in place of experimental facilities.

**Comment:**

Furthermore, given uncertainty over the initial state of the topography (cf. 10 m initial DEM), the uncertainty of deposit depth estimation may be as large or larger than the location errors cited in this study.

**Response:**

We have made some changes to text in places in response to other reviewers to clarify we are not comparing deposit depths in this assessment.

Esposti Ongaro, T., Cerminara, M., Charbonnier, S. J., Lube, G., & Valentine, G. A. (2020). A framework for validation and benchmarking of pyroclastic current models. Bulletin of Volcanology, 82, 1-17.